# Effect of Inappropriate Treatment in Hospitalized Patients with Pyelonephritis Treated with Cefuroxime: A Cohort Study

**DOI:** 10.3390/antibiotics13030274

**Published:** 2024-03-19

**Authors:** Jorge Alberto Cortés, Claudia Rocío Sierra, Ricardo Sánchez

**Affiliations:** 1Department of Internal Medicine, School of Medicine, Universidad Nacional de Colombia, Sede Bogotá, Bogotá 111321, Colombia; 2Infectious Disease Service, Hospital Universitario Nacional, Sede Bogotá, Bogotá 111321, Colombia; 3Laboratorio Clinico y de Patología, Clinica Colsanitas, Grupo de Investigación INPAC, Grupo Keralty, Bogotá 111131, Colombia; 4Clinical Research Institute, School of Medicine, Universidad Nacional de Colombia, Sede Bogotá, Bogotá 111321, Colombia

**Keywords:** urinary tract infections, pyelonephritis, *Escherichia coli*, drug resistance, bacterial, cefuroxime, community-acquired infections, Colombia

## Abstract

The aim of this study was to evaluate the effect of inappropriate therapy in adult patients with community-acquired pyelonephritis caused by *Escherichia coli* receiving empirical treatment with cefuroxime during hospital stay and readmission. A retrospective cohort study was performed. Inappropriate treatment was considered treatment for a nonsusceptible isolate according to the results of the urine culture. Adjustment for confounding factors was performed with propensity score-derived inverse probability of treatment weighting. Between 2013 and 2020, 747 patients were included, 102 (13.7%) of whom received inappropriate therapy. Compared to appropriate therapy, inappropriate therapy was associated with a shorter length of stay in the adjusted analysis (Hazard Ratio = 0.34; 95% CI = 0.23–0.49). After 735 patients were discharged from the hospital, 66 were readmitted in the following 30 days. In comparison with appropriate therapy, inappropriate antimicrobial therapy was not related to readmission (OR 1.47; 95% CI = 0.35–2.79). Inappropriate therapy was not related to a longer hospital stay or readmission due to pyelonephritis after adjusting for confounders and covariates.

## 1. Introduction

Complicated pyelonephritis is a common cause of hospital admission worldwide [1,2] and causes considerable morbidity and costs to healthcare systems. *Escherichia coli* is still the most common isolate and represents more than 50% of microorganisms [3,4]. With hospitalizations rates of between 2.4 and 6.8 per 10,000 individuals for men and between 11.7 and 12.4 per 10,000 individuals for women in the United States and Japan, pyelonephritis represents significant costs for the healthcare systems [5,6], affecting predominantly adult women and men [6].

For the treatment of this pathology, cephalosporins are empirically used widely [3]. Resistance to cephalosporins has increased worldwide among *E. coli* isolates, and resistance mediated by extended-spectrum beta-lactamases (ESBLs) has increased not only in hospital-acquired infections but also in community-acquired infections [7]. Resistance is a widely known factor for failure among infectious diseases; however, its impact on specific drugs in urinary tract infections is now well studied. Data from cohort studies with inappropriate empirical therapy have shown no unfavorable outcomes [8] in some cases and an increased risk of early relapse in others [9].

In this context, the role of second-generation cephalosporins as an option for the treatment of community-acquired pyelonephritis is unclear. The aim of this study was to evaluate the effect of inappropriate empirical therapy with cefuroxime on adult patients hospitalized with community-acquired pyelonephritis in a setting where antibiotic resistance to cephalosporins is common.

## 2. Results

During the study period, 7566 patients with positive urine cultures were identified in the emergency department. Of these, 970 met the inclusion criteria, and 223 met at least one exclusion criterion (Figure 1). Of the 747 patients included, there were 502 women (67.2%) (Table 1). The mean age was 63.8 (SD 19.3) years. Among the women, 370 were menopausal (73.7%), and 9 were pregnant (1.8%). *E. coli* was identified in all isolates and was detected in the blood of 65 patients (8.7%). The susceptibility data are shown in Appendix A. According to susceptibility testing, cefuroxime therapy was considered inappropriate for 102 patients (nonsusceptible, 13.7%). In total, 70 isolates were identified as producers of extended-spectrum beta-lactamases (9.4% of 746 isolates tested).

### 2.1. Patient Characteristics at Admission

Among patients with pyelonephritis, the mean age was 63.8 ± 19.3 years and 32.8% were male. Before IPTW, the patients who had inappropriate therapy were less likely to be women, more likely to be menopausal or older, and more likely to have heart failure (Table 1). No significant differences were observed in the C-reactive protein (CRP) level, creatine level, or frequency of systemic inflammatory response syndrome (SIRS) at admission. The characteristics and differences between the two groups before and after weighting are shown in Table 1. Weighting decreased the differences between the groups (Figure 2). 

### 2.2. Treatment and Outcomes

The median duration of treatment for the entire cohort was 11 days (interquartile range, 8–14 days), with a median of 13 days (10–15.8 days) in the inappropriate therapy group and 11 days in the appropriate therapy group (8–14 days, *p* < 0.001). A total of 102 patients had a change in antimicrobial regimen: 82 in the inappropriate therapy group (80.4%) and 20 in the appropriate therapy group (4.8%, RR 16.7, 95% CI: 11.7–23.9). In the inappropriate therapy group, the antibiotic was changed in 82 cases (80.4%), while it was changed in 31 cases (4.8%) of appropriate therapy. An antibiotic change was carried out most frequently because of the antimicrobial susceptibility testing result. In the inappropriate group, cefuroxime was changed to ertapenem in 67 cases (65.7%), to ceftriaxone in 6 cases (5.9%), and ciprofloxacine in 5 cases (4.9%). Ertapenem was used in in six patients receiving appropriate therapy (0.9%, *p* < 0.001). At discharge, 66 patients who received inappropriate therapy received outpatient parenteral antimicrobial therapy (64.7%), in contrast to 28 patients who received appropriate therapy (4.3%, RR 14.9, 95% CI: 10.1–22.0).

In total, the median length of hospital stay was 4 days (IQR: 3–5.5 days). A total of 11.4% of the patients had a stay of more than 7 days. The median hospitalization time among patients who received inappropriate therapy was 5 days (IQR: 5–6 days), while that among patients who received appropriate therapy was 4 days (IQR: 4–4 days; *p* = 0.021). The frequency of admission to the ICU among patients who received inappropriate therapy (1.0%) was lower than that among patients who received appropriate therapy (6.8%, *p* = 0.038). During hospitalization, 12 patients (1.6%) died, all of whom were in the appropriate therapy group (1.9% vs. 0%, *p* = 0.335).

Hospital readmission due to pyelonephritis was identified in 66 individuals (9.0% of those who were discharged). The frequency was 8.8% in the inappropriate treatment group and 9.0% in the appropriate therapy group (*p* = 1.0). No differences were observed in the median time to readmission between the two groups: 17 days (95% CI: 13–30 days) for the group receiving inappropriate therapy vs. 18 days (95% CI: 14–20 days) for the group receiving appropriate therapy (*p* = 0.7).

### 2.3. Associations between Inappropriate Therapy and Length of Stay and Readmission

In the adjusted regression, inappropriate therapy was not associated with a prolonged length of stay according to a multivariate parametric model (Weibull). The results of this model are shown in Table 2. Oher risk factors identified for a prolonged length of stay included the probability of using an oral treatment, antimicrobial change, and ICU admission. Similar results were obtained using a propensity score-matched subset of the discharge patients, and the same trend was observed for the same variables according to the Poisson model.

In the adjusted regression, inappropriate therapy was not associated with a significantly greater rate of readmission due to pyelonephritis within 30 days after discharge (OR 1.47; 95% CI = 0.75–2.79). Using a propensity score-matched subset, there was also no significant difference in the rate of readmission.

## 3. Discussion

This retrospective cohort study at a tertiary institution in a country with high levels of antibiotic resistance showed that the empirical use of cefuroxime in cases of inappropriate therapy in hospitalized patients with *E. coli* pyelonephritis did not prolong hospitalization time or increase the possibility of readmission when adjusted for other covariates.

There is controversy about the impact of microbiologically inappropriate treatments in the scientific literature. In most studies among patients with urinary tract infections, inappropriate therapy was not associated with worse clinical outcomes. A retrospective cohort study at a Canadian center showed that even in patients with urinary tract infection with bacteremia, the use of empiric therapy was not associated with increased mortality [10]. Another multicenter study that included hospitalized patients from different countries in Europe and the Middle East did not identify inappropriate therapy as a risk factor for therapeutic failure, including mortality, among patients with complicated urinary tract infections [8]. Notably, inappropriate therapy had a protective effect in the study, which was also observed throughout the sensitivity analysis, possibly because patients at lower risk were discharged without changing antibiotics, regardless of susceptibility. In patients infected by multidrug-resistant microorganisms, such as those that produce extended-spectrum beta-lactamases, this resistance pattern seems not to have a deleterious effect on urinary tract infections [11,12].

On the other hand, a recent publication of a secondary analysis of randomized clinical trials for urinary tract infections submitted to the FDA showed an increase in late clinical failure among those with microbiological persistence [9]. However, this study did not include information about the class of antibiotic used for the comparisons. Studies conducted with quinolones have shown that inappropriate therapy may affect the efficacy of these antimicrobial agents [13,14], with a relative higher risk of recurrence. Our study showed a similar rate of readmission secondary to a relapsing or recurring episode of pyelonephritis, but since this was a retrospective study, an information bias could be responsible for some missing endpoints (i.e., patients that were hospitalized in other hospitals that were not able to be identified as recurrent infection).

The objective of an in vitro antimicrobial susceptibility test of the identified microorganism in the laboratory is to provide useful information to clinicians to predict the outcome of patients who receive antimicrobial treatment, usually empirical, and to allow adjustments when decreased susceptibility is identified. In the specific case of urinary tract infections, there is an important limitation in extrapolating this information because the concentration of various antibiotics used for treatment is usually considerably greater in kidney tissue and urine than in blood, as is the case for cefuroxime [15]. A small pharmacokinetic study from the 1980s showed that drug concentrations in urine were considerably greater than the minimum inhibitory concentration and were detected even after 12 h [16]. Furthermore, the urine elimination of cefuroxime occurs after 24 h of administration, and it has an almost linear clearance, with lower clearance rates in the elderly, which is related to renal function [17]. Our study suggests that, with the current cutoff points established for cefuroxime against *E. coli*, it is not possible to predict such outcomes. Larger studies with a precise estimation of the minimal inhibitory concentration might be required to better understand the pharmacokinetic/paharmacodynamic relationship of cefuroxime in patients with urinary tract infections.

Our study identified several factors that affect hospital stay after patient admission, such as changes in antimicrobial therapy use and the availability of oral treatment. In our study, the identification of inappropriate therapy led clinicians to use intravenous drugs such as ertapenem or ceftriaxone, which limits the possibility of an easier discharge and can lead to delays due to administrative problems. The need for intravenous antibiotics can lead to frequent delays in hospital discharge [18], as well as increases in related costs [19]. The results of our study suggest that this is an unnecessary behavior for the majority of patients and might lead to extensive exposure to wider spectrum antibiotics, such as carbapenems, in patients at high risk of readmission.

Our study has several advantages: it involved a more uniform cohort of patients than is usually identified in the literature, taking into account that all patients had *E. coli* infection and that the exposure corresponded to a single antibiotic, which allowed us to extrapolate our information to this group of patients. Additionally, a comprehensive analysis was performed to determine the role of inappropriate therapy in a larger cohort than that found in the literature. The use of IPTW and propensity score-matching favored nonbiased results [20]. Sensitivity analysis performed with propensity score-matched models and other models yielded similar results, providing added strength when considering study risk of bias [21].

However, this study has several limitations. First, this was a retrospective study that was performed at a single institution. Decisions on the use of antimicrobial agents, the time of adjustment and the time of discharge depend on the treating physician, who manages them according to local use. Information biases could have been present, taking into account that it is possible that not all patients with readmission were correctly identified. An attempt was made to reduce the risk of information bias by using standardized definitions, avoiding the use of clinical records with incomplete information and identifying readmissions through the insurer’s audit, making it easier to find all or most of the patients.

## 4. Materials and Methods

### 4.1. Data and Subjects

A retrospective cohort study was performed at Clínica Reina Sofia, a three-level complex care hospital in Bogotá, Colombia. We included information from adult patients hospitalized with a positive urinary tract culture for *E. coli* collected in the emergency department with a clinical diagnosis of pyelonephritis between 1 January 2013 and 30 June 2020. For the clinical diagnosis of pyelonephritis, the presence of symptoms (fever, abdominal or lumbar pain, inflammatory response, or neurological alteration) and a positive urinary culture was considered. Institutional guidelines recommend the use of cefuroxime for the treatment of such patients, and only patients who received this antibiotic as the first choice were included. Patients were excluded if there was an incomplete clinical record, the antibiotic was changed in the first 24 h, they used a permanent or transient urinary catheter, had had previous surgery of the urinary tract, had received immunosuppressive medications (high doses of prednisone or other), had a hematologic malignancy, had a previous episode in the last month, or had previous use of antibiotics in the 30 days preceding the hospitalization event. Susceptibility testing was not performed against cefuroxime or if no outcome data were available because of transfer to another hospital. Patients were identified by use of the Whonet system (ver 5.0, WHO, Geneve, Switzerland), and electronic medical records (EMRs) were reviewed for inclusion and exclusion. Patients were followed up until hospital death or discharge. Data on readmission were searched in the same EMR and in other hospitals through the private insurance network for the following 30 days after discharge (Colsanitas, Keralty, Colombia).

### 4.2. Exposure

The exposure was inappropriate antibiotic therapy, defined on the basis of a nonsusceptibility result on the urine culture against cefuroxime. Cefuroxime was routinely tested during the study period by use of the Vitek XL system (BioMérieux, Marcy-lÉtoile, France) and interpreted using the Clinical Laboratory Standards Institute (CLSI) [22]. During the whole study period the same interpretation rules for susceptibility were used since CLSI did not change them.

### 4.3. Outcomes

The primary outcome was the length of stay, defined as the time from emergency department admission to hospital discharge, which could be ambulatory care or hospital care at home. Secondary outcomes included readmission because of a recurrent event of pyelonephritis and the time to readmission.

### 4.4. Covariates

Selected variables that could be considered confounders were identified in the literature [23,24,25,26]. Briefly, information about certain comorbidities that might affect the risk of a complicated urinary tract infection or the outcome was sought via EMRs. These factors included age, sex, menopause status, urologic disorders such as previous surgery, the presence of transient prosthetic devices, urolithiasis, the presence of immunosuppression caused by pharmacologic interventions, solid or hematologic cancer, human immunodeficiency virus infection or immunologic disorders, previous use of antimicrobial agents in the previous three months, diabetes, chronic renal failure, chronic pulmonary obstructive disease, cirrhosis, or heart failure. We also considered several laboratory measurements within 24 h of admission, including creatinine levels, C-reactive protein levels, leukocyte, and platelet count. The Charlson Comorbidity Index (CCI) was calculated utilizing all the information available, and systemic inflammatory syndrome was identified based on worsening vital signs identified in the first 24 h after admission. Data on the previous use of antibiotics were collected for the previous 90 to 30 days before the hospitalization event.

### 4.5. Statistical Analysis

The characteristics of the patients at admission were evaluated using descriptive statistics. The means and standard deviations or medians and interquartile ranges were calculated as appropriate for continuous data. Discrete data are reported as frequencies and percentages. Unadjusted comparisons were performed using the chi-square test, Fisher’s exact test, or Wilcoxon’s rank-sum test, as appropriate.

The primary analysis was performed by means of an inverse probability of treatment weighting (IPTW) that allowed us to control for confounding factors [27]. We constructed a logistic regression model on the probability of inappropriate therapy. With the propensity score calculated for each individual, the stabilized weights were assigned. A pseudopopulation was constructed for comparison and subsequent analyses. We calculated standardized mean differences (SMDs) in the original and weighted samples to assess covariate balance, and the models were evaluated by examining the balance obtained after the propensity score and the assignment of the weights of the inverse probability of the treatment. The model that achieved the greatest number of variables with SMDs less than 0.1 was chosen [28]. For the primary outcome (length of stay), comparisons were initially made using a survival model, where the outcome variable was the time to hospital discharge and one of the predictor variables was inappropriate therapy. Since the proportional hazards assumptions were violated, parametric models (log log, log normal, exponential, Weibull) were evaluated to adjust for covariates. The model with the best goodness of fit defined by the area under the curve and the Akaike information criterion was chosen. Additional variables related to events that occurred after admission were included in the final model. For the outcome of recurrence, a multiple logistic regression model was performed using the recurrence outcome and taking into account inappropriate therapy as a predictor variable to evaluate the adjustment of changes or variables that were made after admission. The variables were chosen according to the considerations already made, and only because the outcome on this occasion was recurrence.

### 4.6. Sensitivity Analysis

For the sensitivity analysis, for the outcome of length of stay, the propensity score matching of patients who had inappropriate vs. those who had appropriate therapy was performed using a caliper of 0.2 of the logit standard deviation. A similar survival model was used. To confirm the results obtained with IPTW with a survival model, another analysis was performed using a counting model for the pseudo-population.

## Figures and Tables

**Figure 1 antibiotics-13-00274-f001:**
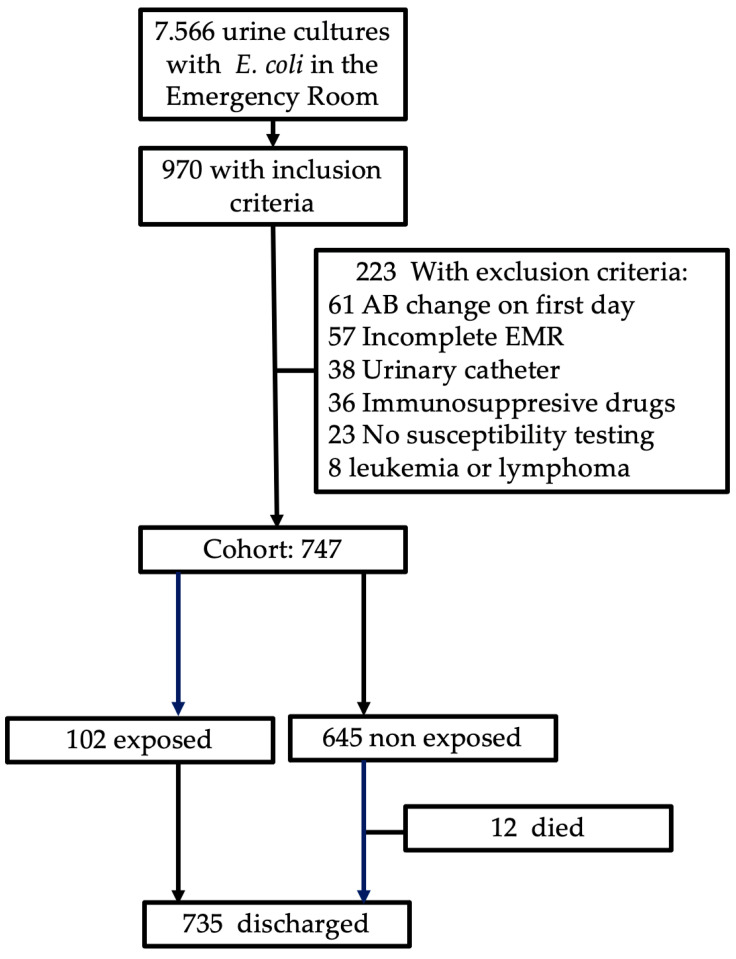
Flow diagram of patients included in the cohort. AB: antibiotic; EMR: electronic medical record.

**Figure 2 antibiotics-13-00274-f002:**
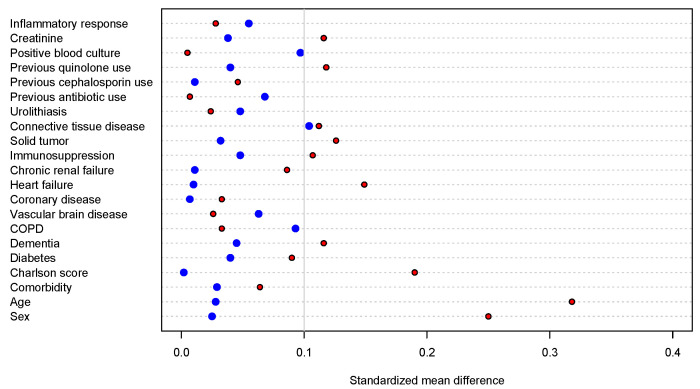
Standardized mean differences between the groups of exposures (inappropriate vs. appropriate therapy) for the full eligible cohort (red) and the inverse probability of treatment weighting propensity score-matched cohort (blue).

**Table 1 antibiotics-13-00274-t001:** Patient characteristics on the date of hospitalization for community-acquired pyelonephritis, by group of exposure (inappropriate therapy), and before and after applying inverse probability weighting.

	Original Sample (N = 747)			After IPTW		
	Inappropriate (N = 102)	Appropriate (N = 645)	SMD	Inappropriate	Appropriate	SMD
Age, years, mean (SD)	68.7 (15.3)	63.0 (19.8)	0.318	63.3 (20.3)	63.8 (19.3)	0.028
Female, n (%)	58 (56.9)	444 (68.8)	0.250	71 (68.4)	434 (67.2)	0.025
Menopause, n (%)	53 (51.9)	317 (49.1)	0.531	53 (51.8)	320 (49.5)	0.021
Any comorbidity, n (%)	55 (53.9)	361 (56.0)	0.041	58 (55.8)	359 (55.7)	0.002
Charlson Comorbidity Index score, mean (SD)	3.6 (1.8)	3.9 (1.7)	0.190	3.9 (1.7)	3.9 (1.7)	0.002
Diabetes, n (%)	22 (21.6)	116 (18.0)	0.090	21 (20.1)	120 (18.5)	0.040
Solid tumor or autoimmune disease, n (%)	17 (16.7)	83 (12.9)	0.107	12 (11.8)	86 (13.3)	0.048
Urolithiasis, n (%)	12 (11.8)	81 (12.6)	0.037	11 (10.8)	80 (12.3)	0.048
Chronic pulmonary diseases, n (%)	9 (8.8)	51 (7.9)	0.033	11 (10.3)	49 (7.6)	0.093
Heart failure, n (%)	7 (6.9)	23 (3.6)	0.192	4 (3.9)	26 (4.1)	0.010
Antibiotic use						
Previous antibiotic use, n (%)	19 (18.6)	122 (18.9)	0.007	17 (16.4)	123 (19.0)	0.068
Previous cephalosporin use, n (%)	7 (6.9)	37 (5.7)	0.046	6 (5.6)	38 (5.8)	0.011
Previous quinolone use, n (%)	6 (5.9)	22 (3.4)	0.118	4 (4.3)	23 (3.5)	0.04
Clinical and laboratory data						
SIRS *, n (%)	59 (57.8)	382 (59.2)	0.028	58 (56.4)	381 (59.1)	0.055
Creatinine mg/dL, mean (SD), n = 730	1.16 (0.48)	1.10 (0.56)	0.116	1.09 (0.43)	1.11 (0.57)	0.038

* Abbreviations: SMD: standardized mean difference; COPD: chronic obstructive pulmonary disease; SIRS: systemic inflammatory response syndrome; CRP: C-reactive protein. Data presented as absolute numbers and proportions, except otherwise marked.

**Table 2 antibiotics-13-00274-t002:** Associations between different variables and length of stay among patients with community-acquired pyelonephritis in a multivariate time to event analysis (Weibull).

Variable	HR *	95% CI
Inappropriate therapy	0.34	0.23–0.49
Oral treatment at discharge	0.22	0.16–0.30
Antimicrobial regimen change	1.86	1.30–2.65
ICU admission	2.63	1.83–3.76

* HR: hazard ratio.

## Data Availability

Anonymized data are available on request.

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
