# Peer review of "Effect of Inappropriate Treatment in Hospitalized Patients with Pyelonephritis Treated with Cefuroxime: A Cohort Study"

_antibiotics, 2024, doi:10.3390/antibiotics13030274_

Round 1

Reviewer 1 Report (New Reviewer)

Comments and Suggestions for Authors

It's an interesting study, but I think that some points should be clarified.

1. What was the treatment duration in the two groups with appropriate antibiotics, as some patients of the inappropriate group received finally the "correct" antibiotic after the results of the cultures?

2. Patients with asymptomatic bacteriuria or cystitis were excluded from the study?

Author Response

It's an interesting study, but I think that some points should be clarified.

  1. What was the treatment duration in the two groups with appropriate antibiotics, as some patients of the inappropriate group received finally the "correct" antibiotic after the results of the cultures?

Answer: The treatment had a median duration of 11 days for the all the patients (interquartile range - IQR 8-14). A median duration of 13 days (10-15.8 days) in the inappropriate therapy group and 11 days in the appropriate therapy group (8-14 days, p<0.001) was observed (lines 80-82).

  1. Patients with asymptomatic bacteriuria or cystitis were excluded from the study?

Answer: Yes, patients with asymptomatic bacteriuria or cystitis were excluded from the study. The inclusion criteria was a to have a pyelonephritis (per medical record), and symptoms.

Reviewer 2 Report (New Reviewer)

Comments and Suggestions for Authors

The presented data is a retrospective single-centre study of inappropriate treatment effect in hospitalized patients with pyelonephritis caused by E. coli from tertiary hospital from Bogotá, Columbia. Authors showed statistical evaluation of patients’ outcome treated empirically by cefuroxime both appropriate or inappropriate according to later antibiotic susceptible testing. The paper is well structured and have sufficient background. The published results were statistically evaluated and the results was partially discussed in Discussion Part. Nevertheless, the text contains a lot of information, which are not well explained or did not give me sense. By my opinion the text and data need to be improved.

Here is mine comments and suggestions:

Line 61 – the percentage of ESBL was calculated from 746 isolates, but included patients were 747, please explain this discrepancy

Tables and figures – the lack of information was in tables or figure description. Sometimes it was hard for me to understand of information and numbers. Please enhance these descriptions. For reader must be presented data easy to understand fully from the table/figure and its description without needing to search the explanation in whole text.

In table 1 – the part about the previous antibiotic use. I have a problem with this part. By my opinion there is no explanation what the “previous” mean. How long is this back side period? By my opinion this, if it means the period tightly before hospital admission, it should be exclusion criterion, because previous antibiotic treatment has undoubtedly affected the patient outcome.

Paragraph 2.2. – this whole paragraph needs to be improved. There is no definition which antibiotic were used, how long the patients were on i.v. treatment. In the text (line 85-87) authors said that 102 patients had change of antibiotic during treatment and 20 of them from appropriate antibiotic, explain why was than antibiotic changed. And on the other hand, if I understand correctly and 102 patients had incorrect therapy according to susceptible test, it means that 20 patients from this group had not changed therapy despite the resistant bacterial strain? Please rewrite and explain.

Line 154-155 – this statement should be very misleading, because I understood this sentence that in case of E. coli the susceptibility test is not needed, because cefuroxime is in all cases clinically effective. explain and discuss - it is due to no infection? or due to inappropriate culture results? Because if you have infection and causative bacteria is resistant to used antibiotic, the administration of antibiotic for which is pathogen resistant could not lead to improve patient’s condition

Do you think that it is due to increase concentration in urine as mentioned later? If yes, it must be improved by other data.

Line 203 – no definition of “clinical diagnosis of pyelonephritis”

The yellow highlighted text – I did not understand what it means

Comments on the Quality of English Language

The presented data is a retrospective single-centre study of inappropriate treatment effect in hospitalized patients with pyelonephritis caused by E. coli from tertiary hospital from Bogotá, Columbia. Authors showed statistical evaluation of patients’ outcome treated empirically by cefuroxime both appropriate or inappropriate according to later antibiotic susceptible testing. The paper is well structured and have sufficient background. The published results were statistically evaluated and the results was partially discussed in Discussion Part. Nevertheless, the text contains a lot of information, which are not well explained or did not give me sense. By my opinion the text and data need to be improved.

Here is mine comments and suggestions:

Line 61 – the percentage of ESBL was calculated from 746 isolates, but included patients were 747, please explain this discrepancy

Tables and figures – the lack of information was in tables or figure description. Sometimes it was hard for me to understand of information and numbers. Please enhance these descriptions. For reader must be presented data easy to understand fully from the table/figure and its description without needing to search the explanation in whole text.

In table 1 – the part about the previous antibiotic use. I have a problem with this part. By my opinion there is no explanation what the “previous” mean. How long is this back side period? By my opinion this, if it means the period tightly before hospital admission, it should be exclusion criterion, because previous antibiotic treatment has undoubtedly affected the patient outcome.

Paragraph 2.2. – this whole paragraph needs to be improved. There is no definition which antibiotic were used, how long the patients were on i.v. treatment. In the text (line 85-87) authors said that 102 patients had change of antibiotic during treatment and 20 of them from appropriate antibiotic, explain why was than antibiotic changed. And on the other hand, if I understand correctly and 102 patients had incorrect therapy according to susceptible test, it means that 20 patients from this group had not changed therapy despite the resistant bacterial strain? Please rewrite and explain.

Line 154-155 – this statement should be very misleading, because I understood this sentence that in case of E. coli the susceptibility test is not needed, because cefuroxime is in all cases clinically effective. explain and discuss - it is due to no infection? or due to inappropriate culture results? Because if you have infection and causative bacteria is resistant to used antibiotic, the administration of antibiotic for which is pathogen resistant could not lead to improve patient’s condition

Do you think that it is due to increase concentration in urine as mentioned later? If yes, it must be improved by other data.

Line 203 – no definition of “clinical diagnosis of pyelonephritis”

The yellow highlighted text – I did not understand what it means

Author Response

The presented data is a retrospective single-centre study of inappropriate treatment effect in hospitalized patients with pyelonephritis caused by E. coli from tertiary hospital from Bogotá, Columbia. Authors showed statistical evaluation of patients’ outcome treated empirically by cefuroxime both appropriate or inappropriate according to later antibiotic susceptible testing. The paper is well structured and have sufficient background. The published results were statistically evaluated and the results was partially discussed in Discussion Part. Nevertheless, the text contains a lot of information, which are not well explained or did not give me sense. By my opinion the text and data need to be improved.

Here is mine comments and suggestions:

  1. Line 61 – the percentage of ESBL was calculated from 746 isolates, but included patients were 747, please explain this discrepancy

Answer: The ESBL result was omitted in a report from one patient.

  1. Tables and figures – the lack of information was in tables or figure description. Sometimes it was hard for me to understand of information and numbers. Please enhance these descriptions. For reader must be presented data easy to understand fully from the table/figure and its description without needing to search the explanation in whole text.

Answer: More information of the figures and tables were added.

  1. In table 1 – the part about the previous antibiotic use. I have a problem with this part. By my opinion there is no explanation what the “previous” mean. How long is this back side period? By my opinion this, if it means the period tightly before hospital admission, it should be exclusion criterion, because previous antibiotic treatment has undoubtedly affected the patient outcome.

Answer: The reviewer is right. The information about previous use of antibiotics was collected for the 90 days before the hospitalization. A previous event of pyelonephritis or antibiotic use in the previous 30 days were exclusion criteria. The information was added in the methodology section.

  1. Paragraph 2.2. – this whole paragraph needs to be improved. There is no definition which antibiotic were used, how long the patients were on i.v. treatment. In the text (line 85-87) authors said that 102 patients had change of antibiotic during treatment and 20 of them from appropriate antibiotic, explain why was than antibiotic changed. And on the other hand, if I understand correctly and 102 patients had incorrect therapy according to susceptible test, it means that 20 patients from this group had not changed therapy despite the resistant bacterial strain? Please rewrite and explain.

Answer: The information was provided and the paragraph was changed. The reason for the change of antibiotic was the availability of the results of antimicrobial susceptibility testing.

  1. Line 154-155 – this statement should be very misleading, because I understood this sentence that in case of E. coli the susceptibility test is not needed, because cefuroxime is in all cases clinically effective. explain and discuss - it is due to no infection? or due to inappropriate culture results? Because if you have infection and causative bacteria is resistant to used antibiotic, the administration of antibiotic for which is pathogen resistant could not lead to improve patient’s condition. Do you think that it is due to increase concentration in urine as mentioned later? If yes, it must be improved by other data.

Answer: We retired the statement. In the following paragraph the idea is better developed to suggest what the reviewer is suggesting. Among some explanations for our results is the possibility the current breakpoints are not accurate for cefuroxime and that could be related to the increased concentration of the drug in the renal tissue.

  1. Line 203 – no definition of “clinical diagnosis of pyelonephritis”

Answer:  The definition was added.

  1. The yellow highlighted text – I did not understand what it means

Answer:

  • Lines 32-35: It was changed to reflect the importance of hospitalization and costs.
  • Lines 37-46: It was simplified to express the possible but not well studied effect of resistance among urinary tract isolates.
  • Table 1: It was changed to solid tumors ()instead of oncological diseases)
  • Lines 74-80: It is a description of the baseline characteristics of the patient population. General information was added. It presents the differences between the tow groups before the Inverse Probability Treatment Weighting (IPTW).
  • Line 116-123: It presents the results of the models performed to evaluate the effect of the exposure on the outcomes, after adjusting for the covariates that wer not included in the propensity score. The description of table 2 was also changed.
  • Line 145-155: It was modified to express better the differences between those studies with different results in the literature.
  • Line 184-185: IPTW is a methodology to balance the variables included in cohort studies to diminish the risk of biased results. The line was not changed since we think is an ad antage of the methodology used.
  • Line 207-209: Those patients with high doses of steroids, those with hematological malignancy were excluded by request of a previous peer review that considered that those conditions do not reflect the clinical situation of a community-acquired pyelonephritis.
  • Line 224-225: It was change to clarify that the interpretation of the susceptibility testing did not change during the time (2013-2020).
  • Line 255-256: As stated this methodology allows for a balanced use of data from different groups which adjusts the risk of known confunding factors to influence the outcome.

Reviewer 3 Report (New Reviewer)

Comments and Suggestions for Authors

An interesting manuscript evaluating the effect of inappropriate treatment of pyelonephritis with cefuroxime is presented.

The results are somewhat surprising, because it is reported that inappropriate therapy was not related to a longer hospital stay or readmission due to pyelonephritis.

Unfortunately, I cannot recommend the manuscript for acceptance due to, in my opinion, a completely inadequate methodology in the microbiological part of the manuscript. There is a complete lack of description of the methodology of how Escherichia coli strains were identified, and it is clear that no quality control is provided in the case of antibiotic susceptibility/resistance testing of isolated strains. The text mentions the production of ESBL, but again the methodology of its detection is not mentioned. I am afraid that the results based on the determination of susceptibility/resistance, which is not reliably documented, may not be completely adequate. An adequate assessment would be possible not only on the basis of the qualitative susceptibility/resistance of Escherichia coli strains to cefuroxime, but also on the basis of the analysis of the minimum inhibitory concentrations of cefuroxime (due to the excretion of this antibiotic mainly in the urine). In my opinion, the chosen analysis based only on the qualitative evaluation of susceptibility/resistance is not sufficient to formulate the stated conclusions.

Author Response

An interesting manuscript evaluating the effect of inappropriate treatment of pyelonephritis with cefuroxime is presented.

The results are somewhat surprising, because it is reported that inappropriate therapy was not related to a longer hospital stay or readmission due to pyelonephritis.

Unfortunately, I cannot recommend the manuscript for acceptance due to, in my opinion, a completely inadequate methodology in the microbiological part of the manuscript. There is a complete lack of description of the methodology of how Escherichia coli strains were identified, and it is clear that no quality control is provided in the case of antibiotic susceptibility/resistance testing of isolated strains. The text mentions the production of ESBL, but again the methodology of its detection is not mentioned. I am afraid that the results based on the determination of susceptibility/resistance, which is not reliably documented, may not be completely adequate. An adequate assessment would be possible not only on the basis of the qualitative susceptibility/resistance of Escherichia coli strains to cefuroxime, but also on the basis of the analysis of the minimum inhibitory concentrations of cefuroxime (due to the excretion of this antibiotic mainly in the urine). In my opinion, the chosen analysis based only on the qualitative evaluation of susceptibility/resistance is not sufficient to formulate the stated conclusions.

Answer: We understand the point of the reviewer. However, numerous  papers have been published with the same methodology. Identification of Escherichia coli and, in general, Enterobacterales, with the methodology used has been excellent, with results over 98% for concordant identification using automated methodology (Menozzi et al. J Clin Microbiol 2006 ;44(11):4085-94. doi: 10.1128/JCM.00614-06.). Susceptibility testing for this antimicrobial has not been changed since 2010, and this was the interpretation used for this well known automated system for laboratories, Vitek (France). It is mentioned in line 238 and reference 22. Validation studies for the performance of Vitek system has shown a very good agreement with different cephalosporins, with a categorical agreement over 95% (Bobenchik et al. J Clin Microbio 2015;53(3):816-23. doi: 10.1128/JCM.02697-14).

Similar papers in desgin include those that make an evaluation of quinolones (Benavides et al. Eur J Clin Microbiol & Infect Dis (2022); 41:741–749. https://doi.org/10.1007/s10096-022-04428-1), third generation cephalosporins (Mark et al. Ann Emerg Med. 2021;78:357-369), ESBL (Kim et al. Eur J  Clin Microbiol Infect Dis 2019; 38:937–944. https://doi.org/10.1007/s10096-019-03528-9), cefepime and piperacillin/tazobactam (Branton et al. JAC Antimicrob Resist. 2023 Mar 17;5(2):dlad021. doi: 10.1093/jacamr/dlad02), among others. But it is not only that other investigators have used this retrospective design. In a broader sense, laboratory information has been used for clinicians to take decisions in regard to antibiotic use for individual patients over 50 years, and although the interpretation has some limitations (Dalhoffe et al. Infection 2009; 37: 296–305. DOI 10.1007/s15010-009-7108-9), information for the clinical correlation between breakpoints interpretation and clinical data for this antibiotic is lacking and this study may call for a better definitions of  the breakpoints or increasing the quality of the results provided by the different technologies tha are available. 

Round 2

Reviewer 2 Report (New Reviewer)

Comments and Suggestions for Authors

The made changes in paper led by my opinion to better understanding of text and also the results are more clear now. The paper is prepared for publication.

Reviewer 3 Report (New Reviewer)

Comments and Suggestions for Authors

I am pleased to say that the edited version of the manuscript addresses my comments and reservations about the first version. The authors added information on the assessment of the susceptibility/resistance of Escherichia coli strains to cefuroxime and at the same time on its concentration in urine.

I now consider the manuscript suitable for publication and I am convinced that the text modifications have significantly increased the quality of the article.

This manuscript is a resubmission of an earlier submission. The following is a list of the peer review reports and author responses from that submission.

Round 1

Reviewer 1 Report

Comments and Suggestions for Authors

A well-written text, the English is clear and understandable. However, the subject of the study could have been more interesting. The methodology and findings of the study could have been explained more clearly. The definition of patient group and community-acquired pyelonephritis is not clear. There are patients who were defined as community-acquired pyelonephritis but who had a urinary catheter and prosthetic material inserted. and immunocompromised patients. It falls under the definition of healthcare-associated infection. The statistical analysis is well done. Discussion could have been more detailed.

Reviewer 2 Report

Comments and Suggestions for Authors

- I wonder what is new in your study and results? or what is your unique findings and its importance?

-Punctuation  and typo errors are seen, kindly revise the whole manuscript.

-Escherichia coli term should appear with its well-known abbreviation in the abstract for the first time then only abbreviation will be OK.

- Introduction section is poor and short.

- “The aim of this study is to evaluate the effect of inappropriate empirical therapy with cefuroxime” This appears in the introduction section, however the title of your article seems to be more wide (treatment rather than cefuxime)
-Section 2.1 requires careful language editing

- In line 54: Do you mean creatine or creatinine?

- In tabe1: regarding age, what is shown? Mean and SD?

- Again revise the abbreviation CRP and its term

-You mention that you study the basic hematologic data, what do you mean? Can you clarify this in the methods and in results sections.

- Inappropriate treatment, mention, in details, the criteria of inappropriate

- Line 148-150: what the significance of this part (in vitro study results)?

Comments on the Quality of English Language

Language editing is essential, many structural, typo, spacing errors are seen